# Mental health and well-being among Ukrainian female university students: The impact of war over 3 years

Alexander Reznik[1], Iuliia Pavlova[2] , Valentyna Pavlenko[3], Anton Kurapov[4], Alexander Drozdov[5], Nataliia Korchakova[6], Shai-li Romem Porat[1] and Richard Isralowitz[1]

[1]Regional Alcohol and Drug Abuse Research (RADAR) Center, Ben-Gurion University of the Negev, Israel; [2]Department of Theory and Methods of Physical Culture, Lviv State University of Physical Culture named after Ivan Bobersky, Ukraine; [3]Department of Applied Psychology, V N Karazin Kharkiv National University, Ukraine; [4]Department of Experimental and Applied Psychology, Taras Shevchenko National University of Kyiv, Ukraine, Ukraine; [5]General, Developmental & Pedagogical Psychology, T H Shevchenko National University Chernihiv Colehium, Ukraine and [6]Department of Developmental & Pedagogical Psychology, Rivne State University of Humanities, Ukraine

## Rapid Communication

**Keywords:**
women; war; mental health; Ukraine

**Corresponding author:**
Richard Isralowitz;
Email: richard@bgu.ac.il

## Abstract

The Russia-Ukraine war generates fear, depression, loneliness, burnout and substance misuse among civilians. Our study examines mental health among Ukrainian university female students during 3 years of war. A total of 3,467 students were surveyed on three occasions: August to October 2022 (T1, $n = 1,416$), March to July 2023 (T2, $n = 747$) and September to November 2024 (T3, $n = 1,304$). The respondent's average age was 19.3 years, 25.3% identified as secular and 36.9% were married/partnered. The respondents included 81.2% who were not relocated, 10.7% who were internally displaced and 8.1% who were refugees. Valid and reliable survey instruments were used to gather data. One-way analysis of variance (ANOVA) shows a significant decrease in fear of war, depression and burnout in 2023 compared to 2022; however, there was a marked increase in 2024. Regardless of the survey period, one-way ANOVA shows a significant difference in fear of war and burnout scores associated with depression and loneliness levels. Stepwise regression analysis shows fear of war, depression and loneliness associated with burnout. This study provides usable information for mental health services planning and intervention purposes associated with young women affected by war in Ukraine. Additionally, it has relevance for training to address client and service personnel needs, for academic curriculum development and course instruction, and as a reference source for mental health personnel addressing student needs.

## Impact statement

The impact of war on women remains poorly understood, especially in the long-term perspective. Young college-aged women are particularly vulnerable to the impact of war on their mental health and well-being. Over the 3 years of war (2022–2024), their values for fear of war, depression and loneliness varied significantly from year to year. However, it cannot yet be concluded that such changes indicate a lasting improvement in mental health and well-being among female respondents and their adaptation to disaster conditions. This article has implications for informed decision-making related to policies and services to promote the mental health and well-being of women in war-torn environments, in Ukraine and elsewhere, providing key aspects for focused interventions.



## Introduction

The war between Russia and Ukraine began on February 24, 2022, provoking the most serious military conflict in Central Europe since 1945. After nearly 3 years of war, the Office of the United Nations High Commissioner for Human Rights (UNOHCHR) verifies about 13,000 deaths and 32,000 injuries among the Ukrainian civil population. Russian missiles and drones have damaged or destroyed schools, hospitals and energy facilities, including those that provide essential water, gas and electricity. Evacuations of children with families, older adults and disabled people continue in response to shifting areas of conflict. Despite ongoing international efforts to negotiate prisoner exchange and ceasefire, Ukraine continues to experience extensive damage associated with the health and well-being of its civilian population, according to multiple information sources (Hryhorczuk et al., 2024; An et al., 2025; UNOHCHR, 2025; UN Ukraine, 2025).

### War and the health of women

Determination of war impact is complex because of the multiple risk factors and disorders involved, as well as the different measurement methods used over time and location. However, the World Health Organization estimates nearly 9.6 million Ukrainians may have mental health problems, women more than men, including possible conditions of anxiety, post-traumatic stress disorder (PTSD), fear, depression, loneliness, suicidal ideation, alcohol and other harmful substance use (Kurapov et al., 2023a,b; Pavlenko et al., 2023, 2024a; Sheather, 2022; Trends Research & Advisory, 2024; WHO, 2024). Specific to the country's female population, nearly 6.7 million need humanitarian assistance (UNOCHA, 2025), and more than 4,000 females have been killed, with the real death toll likely to be much higher (UN Women, 2025). Studies conducted in Ukraine about the impact of war on the mental health and well-being of the civilian population show women are more at risk of developing psychological disorders than men (Chudzicka-Czupała et al., 2023; Kurapov et al., 2023a; Martsenkovskyi et al., 2024; Zasiekina et al., 2024; Tessitore et al., 2025). Regarding age, depression symptoms among women aged 18–24 years are nearly twice the amount recorded for men (UNOCHA, 2024). A large-scale study of mental health in Ukrainian women shows they are significantly more likely than males (39.0% vs. 33.8%) to have depression, generalized anxiety disorder, PTSD, as well as excessive alcohol and cannabis use (Martsenkovskyi et al., 2024). Another consequence of war is its impact on women's health and reproduction (Tatarchuk et al., 2024; Siusiuka et al., 2025). Gender-based violence, already high before the war, has increased by 36% since 2022, and essential services to address health and mental health needs have deteriorated, making living conditions difficult for women and those under their care, especially in heavily bombed areas (European Commission, 2023; Pavlenko et al., 2024b).

This article describes Ukrainian university female student's physical and mental health during the 3 years of war. Females being most of the survey respondents, along with their high-risk status for health and mental health disorders, are the principal reasons for gender-specific investigation. We hypothesize that female student well-being, defined in terms of mental health conditions, has not changed significantly over the study period, evidencing possible adaptation and resilience to war.

## Method

### Study design

A cross-sectional study was conducted three times to identify potential changes and trends in mental health and well-being among Ukrainian female university students over the 3 years of war.

The survey was conducted online and anonymously using the Qualtrics software platform. Informed consent was contained in the introductory part of the questionnaire. In the case of refusal to participate, an automatic exit from the online survey system occurred. The start of the survey means respondent's informed consent. All participants were over 16 years of age, and informed consent was not required from the parents or legally authorized representatives.

### Instruments

Previous studies (Kurapov et al., 2023b; Pavlenko et al., 2024a) about the impact of war on Ukrainian students and professionals has shown that four variables tend to describe their mental health – fear of war, depression, burnout and loneliness. These factors were used in this study to assess changes in the mental health of female students over the 3-year study period.

Standardized data collection instruments included four data collection scales. The first was the 10-item Fear of War Scale (FWS) (Kurapov et al., 2023b). The agreement levels for the statements used are evaluated by a 5-point Likert scale ranging from 1 (*strongly disagree*) to 5 (*strongly agree*). Higher total scores correspond with more fear of war. Examples of the questions used are "I am scared because war costs human lives," "I am afraid of losing my life because of the war," and "I have a sleep disorder because I worry the war will get to me." Other scales used for this study include the nine item Patient Health Questionnaire (PHQ-9) for measuring the severity of depression (Kroenke et al., 2001). Item 9 of the PHQ-9 ("Thoughts that you would be better off dead or hurting yourself in some way") was used as a dichotomous (yes/no) indicator of suicidal ideation. Also, the 10-item Short Burnout Measure (SBM) (Malach-Pines, 2005) and the De Jong Gierveld 6-item Loneliness Scale, which assesses emotional, social and general loneliness, were used for the survey (Gierveld & Tilburg, 2006).

All instruments were translated from English to Ukrainian and back-translated. Cronbach's alpha scores for the scales used are FWS = 0.849, PHQ-9 = 0.885, SBM = 0.888 and Loneliness Scale = 0.794. Survey respondents provided information about their age, marital status, religiosity (secular/non-secular) and location status (i.e., not relocated, internally relocated or refugee). Statistical analyses were conducted using SPSS, version 29. Stepwise multiple regression, $t$-test, Chi-square test, the Mann–Whitney test, the Kruskal–Wallis test, one- and two-way analysis of variance (ANOVA), as well as effect size measures, were used for the data analysis.

### Participants

A snowball, a non-probability sampling technique was used for online data collection among the Ukrainian female university students. Respondents who did not indicate their gender, university affiliation or affiliation with universities outside of Ukraine (Germany, Poland, etc.) were excluded from the sample. A total of 3,467 female students were surveyed on three occasions: August to October 2022 (T1, $n = 1,416$), March to July 2023 (T2, $n = 747$) and September to November 2024 (T3, $n = 1,304$). Respondent's average age was 19.3 years (median 18.0 years), 25.3% identified as secular and 36.9% were married/partnered. The respondents included 81.2% who were not relocated, 10.7% who were internally displaced and 8.1% who were refugees. The respondents were from six locations (i.e., Kyiv, Kharkiv, Lviv, Chernihiv, Rivne and Ternopil). The contribution of each location varied from year to year. Based on the results of T1…T3, the contribution of each location to the sample was as follows: Lviv 27.1%, Chernihiv 16.3%, Kharkiv 15.3%, Ternopil 15.1%, Rivne 14.6% and Kyiv 8.3%, and students affiliated with universities in other locations 3.3%.

Table 1 provides demographic information about the participants by survey period.

## Results

### Fear of war

For those surveyed, the mean value of the FWS was 34.8 (standard deviation [SD] = 7.2), with a range of 10–50. One-way ANOVA

**Table 1.** Demographic data

| | Total (n = 3,467) | 2022 (n = 1,416) | 2023 (n = 747) | 2024 (n = 1,304) | p-values[a] |
|---|---|---|---|---|---|
| Age, mean (SD) | 19.3 (4.2) | 19.5 (4.3) | 18.9 (2.3) | 19.3 (4.9) | 0.012 |
| Median | 18.0 | 18.0 | 18.0 | 18.0 | |
| Religiosity, n (%) | | | | | |
| Secular | 874 (25.3) | 377 (26.7) | 114 (15.3) | 383 (29.5) | <0.001 |
| Non-secular | 2,581 (74.7) | 1,037 (73.3) | 630 (84.7) | 914 (70.5) | |
| Marital status, n (%) | | | | | |
| Married/partner | 1,268 (36.9) | 528 (37.6) | 283 (38.0) | 457 (35.4) | 0.382 |
| Other | 2,172 (63.1) | 878 (62.4) | 461 (62.0) | 833 (64.6) | |
| Location status, n (%) | | | | | |
| Non-relocated | 2,794 (81.2) | 1,091 (77.6) | 671 (89.9) | 1,032 (79.9) | <0.001 |
| Internally relocated | 368 (10.7) | 147 (10.5) | 33 (4.5) | 188 (14.6) | |
| Refugees | 281 (8.1) | 168 (11.9) | 42 (5.6) | 71 (5.5) | |

[a]p-value of *t*-test and Chi-square test.

shows a significant difference in fear scores associated with survey period ($F_{(2,3,102)} = 18.010$, $p < .001$, partial $\eta^2 = .011$). Regardless of survey period, one-way ANOVA shows a significant difference in fear scores associated with respondents' location status: 34.6 versus 35.1 versus 36.9 among non-relocated, internally relocated and refuge respectively ($F_{(2,3,093)} = 12.355$, $p < .001$, partial $\eta^2 = .008$). T-test shows higher fear scores among married/partnered students ($t_{(3087)} = 2.828$; $p = .002$, $d = 7.191$). The two-way ANOVA reveals significant interaction between T1…T3 and religious status associated with FWS values ($F_{(2,3,097)} = 5.714$; $p = .003$, partial $\eta^2 = .004$).

### Burnout

The mean value of the SBM was 28.6 (SD = 7.7), with a range of 10–50. One-way ANOVA shows a significant difference in SBM scores associated with survey period ($F_{(2,2,789)} = 10.318$, $p < .001$, partial

$\eta^2 = .007$). Also, regardless of T1…T3, one-way ANOVA shows different SBM scores associated with respondents' location status: 28.1 versus 30.6 versus 30.8 among non-relocated, internally relocated and refugees, respectively ($F_{(2,2,781)} = 25.602$, $p < .001$, partial $\eta^2 = .018$). T-test shows higher SBM scores among secular ($t_{(2788)} = 6.553$; $p < .001$, $d = 7.628$) and married/partnered respondents ($t_{(2776)} = 2.140$; $p = .032$, $d = 7.697$).

Figure 1 provides information on the fear of war and burnout scores by T1…T3 periods.

### Depression

The PHQ-9 (i.e., depression severity) mean value was 10.0 (SD = 6.3) with a range of 0–27. One-way ANOVA shows a significant difference in PHQ-9 scores associated with survey periods T1…T3 ($F_{(2,2,772)} = 23.924$, $p < .001$, partial $\eta^2 = .017$). Also, regardless of survey period, one-way ANOVA shows different

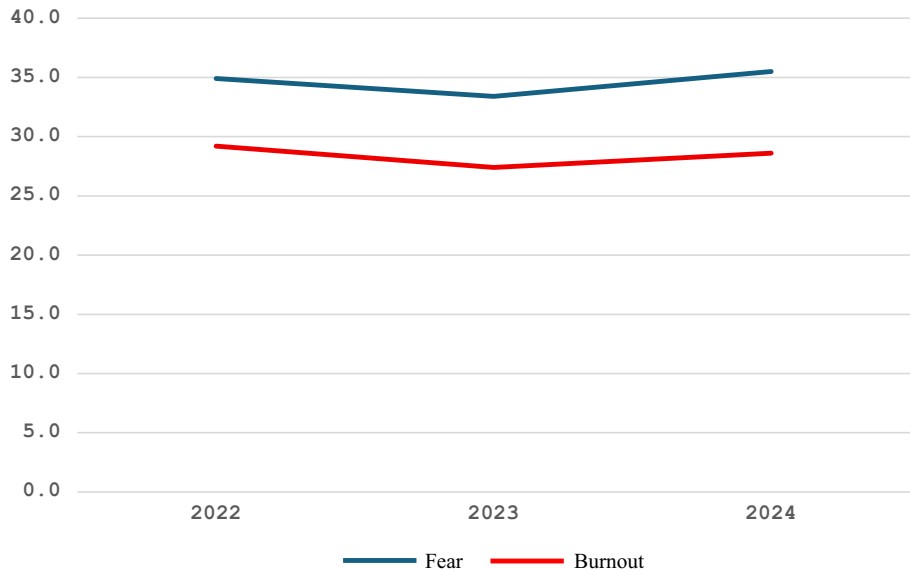

**Figure 1.** Fear of war and burnout by survey period.

PHQ-9 scores associated with respondents' location status: 9.7 versus 11.2 versus 10.6 among non-relocated, internally relocated and refugees, respectively ($F(2,2,764) = 9.169$, $p < .001$, partial $\eta^2 = .007$). T-test shows a higher PHQ-9 score among secular than religious respondents ($t(2771) = 7.606$; $p < .001$, $d = 6.192$).

Figure 2 provides information on the depression score by the T1…T3 periods.

Based on PHQ-9 validation (Kroenke et al., 2001), PHQ-9 scores were divided into five groups: 0–4 (no/minimal depression), 5–9 (mild depression), 10–14 (moderate depression), 15–19 (moderately severe depression) and 20–27 (severe depression). Figure 3 provides information on depression levels by T1…T3 periods.

Regardless of survey period, one-way ANOVA shows a significant difference in FWS and SBM scores associated with depression level: $F(4,2,699) = 49.490$, $p < .001$, partial $\eta^2 = .068$ and $F(4,2,722) = 560.612$, $p < .001$, partial $\eta^2 = .452$, respectively (Figure 4).

Also, regardless of survey period, 29.6% of the respondents reported last month suicidal ideation, secular more than religious (39.2% vs. 26.1%; $\chi^2(1, N = 2,798) = 45.950$, $p < .001$).

### Loneliness

Results of the De Jong Gierveld 6-Item Loneliness Scale were presented on ordinal scales from 0 (*no loneliness*) to 3 (*high loneliness*) for emotional and social loneliness and from 0 (*no loneliness*) to 6 (*very high loneliness*) for general loneliness. Kruskal–Wallis test shows emotional loneliness significantly decreased from T1 to T3 ($H(2, N = 2,737) = 8.048$, $p = .018$, $\eta^2 = .002$). Regardless of the survey period, the Kruskal–Wallis test does not evidence a significant association between emotional, social and general loneliness and location status. Mann–Whitney test shows a higher level of emotional loneliness among secular than religious respondents

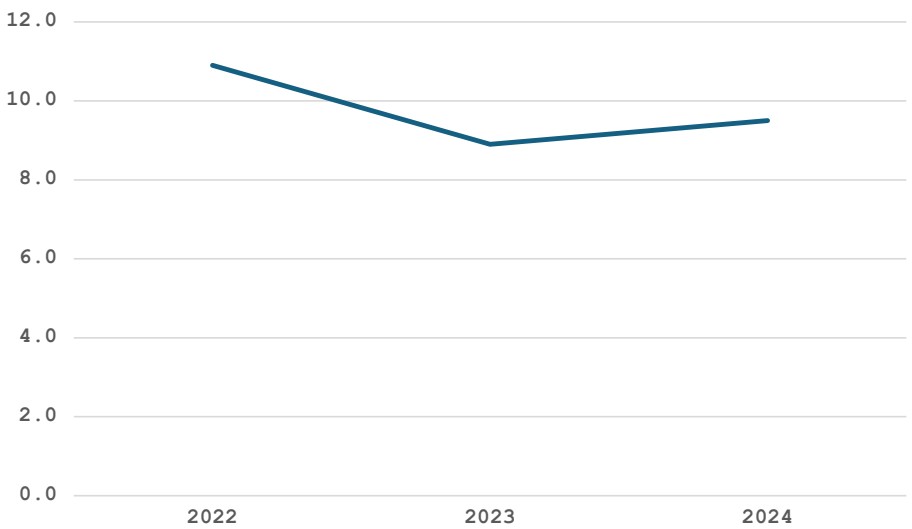

**Figure 2.** Depression by survey period.

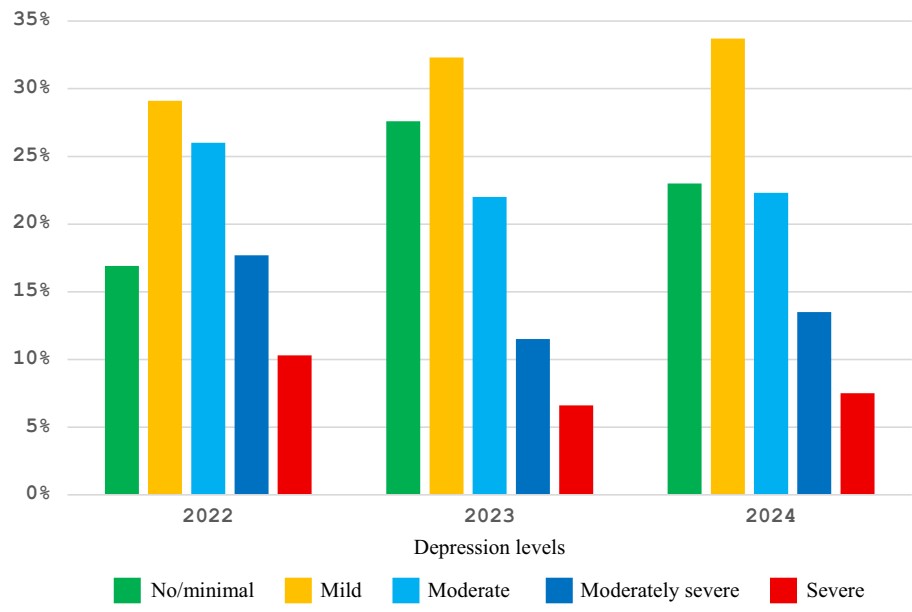

**Figure 3.** Depression level distributions by survey period.

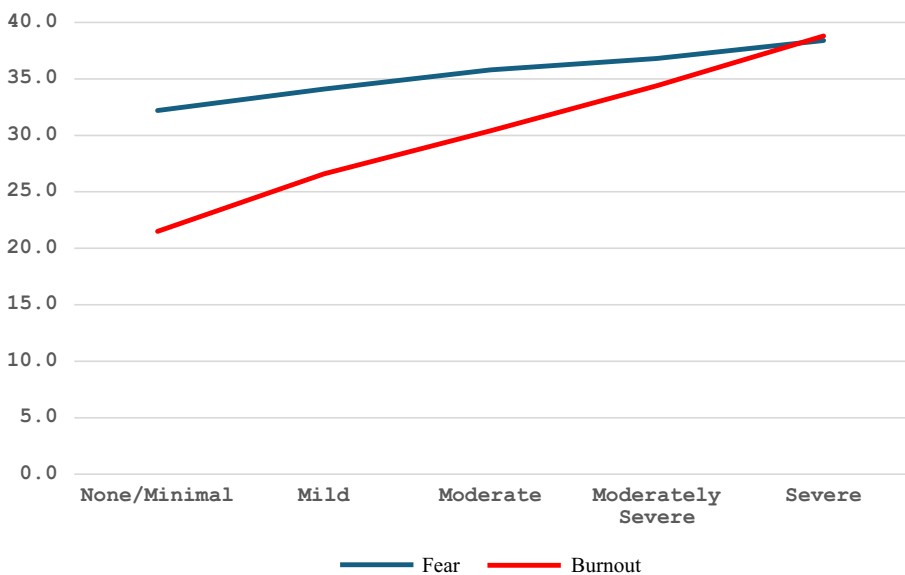

**Figure 4.** Fear of war and burnout by depression level.

($U$ = 701,485.0, $Z$ = -2.311, $p$ = .021, $r$ = .044). Also, married/partnered respondents report lower levels of emotional and general loneliness than those not coupled ($U$ = 779,723.0, $Z$ = -4.547, $p$ < .001, $r$ = .087 and $U$ = 789,191.5, $Z$ = -3.722, $p$ < .001, $r$ = .071, respectively). Regarding depression, Kruskal–Wallis test shows a significant association between increased emotional, social and general loneliness and increased depression: $H(4, N = 2,692)$ = 650.499, $p$ < .001, $\eta^2$ = .255; $H(4, N = 2,700)$ = 121.915, $p$ < .001, $\eta^2$ = .043; and $H(4, N = 2,687)$ = 484.345, $p < .001, \eta^2$ = .179 respectively.

Regardless of survey period, one-way ANOVA shows a significant difference in FWS scores associated with emotional and general loneliness levels: $F(3,2,658)$ = 34.301, $p$ < .001, partial $\eta^2$ = .037 and $F(6,2,646)$ = 11.543, $p$ < .001, partial $\eta^2$ = .026, respectively. Also, one-way ANOVA shows a significant difference in SBM scores associated with emotional, social and general loneliness levels: $F(3,2,683)$ = 307.186, $p$ < .001, partial $\eta^2$ = .256; $F(3,2,687)$ = 37.742, $p$ < .001, partial $\eta^2$ = .040; and $F(6,2,672)$ = 102.102, $p$ < .001, partial $\eta^2$ = .187, respectively (Figure 5).

Two-way ANOVA evidences a significant difference in FWS scores based on survey period with emotional and general loneliness: $F(6,2,650)$ = 3.442, $p$ = .002, partial $\eta^2$ = .008 and $F(12,2,632)$ = 2.913, $p$ < .001, partial $\eta^2$ = .013, respectively.

Table 2 represents stepwise regression analysis results for burnout. The proportion of variation (i.e., adjusted $R^2$) for burnout predicted by these variables is 0.563. Additional independent variables, such as age and religion, do not significantly increase the proportion of explained variance.

### Discussion and conclusion

Purposive samples of Ukrainian university female students were compared in terms of their fear of war, depression and loneliness in war conditions. The present results are consistent with other studies conducted on the impact of war in Ukraine (b; Pavlenko et al., 2023; Kurapov et al., 2023a,b, 2024; Pavlenko et al., 2024a) and elsewhere

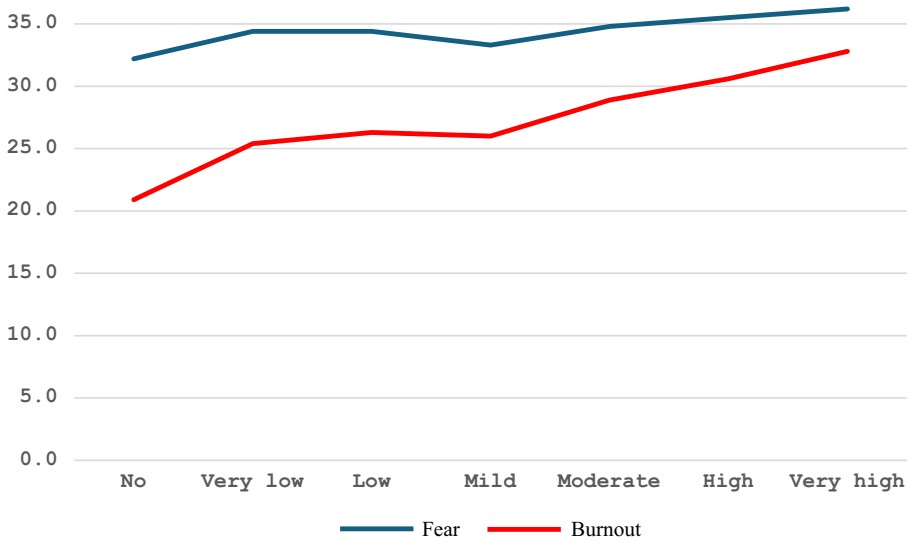

**Figure 5.** Fear of war and burnout due to the general loneliness level.

**Table 2.** Burnout – stepwise regression

| Variable | B | 95% CI for B | | SE B | β |
|---|---|---|---|---|---|
| | | LL | UL | | |
| Constant | 6.125 | 4.277 | 7.973 | .943 | |
| Depression | .682 | .647 | .718 | .018 | .557*** |
| Fear of war | .243 | .218 | .268 | .013 | .315*** |
| Loneliness | .610 | .492 | .729 | .060 | .147*** |
| T3[a] | 2.325 | 1.729 | 2.742 | .258 | .142*** |
| Non-relocation[a] | 1.238 | .746 | 1.729 | .250 | .065*** |
| Marital status[b] | −.715 | −1.125 | −.304 | .209 | −.045*** |

*Note*: CI, confidence interval; LL, lower limit; UL, upper limit.
[a] 1 = yes, 2 = no.
[b] 1 = married, 2 = non-married.
***$p < .001$.

(Carpiniello, 2023; Conflict Watchlist, 2024; M. Ahmed et al., 2024; Our World in Data, 2024; The Peace Research Institute Oslo, 2024). However, there is a dearth of information about the mental health and well-being of women examined at three time periods of war between 2022 and 2024.

The present results partially support the study hypothesis. The values of fear of war, depression and loneliness changed significantly during the three study periods. However, for now, it is not possible to claim that such changes evidence lasting mental health and well-being improvement among the female respondents. The results show a significant decrease in the fear of war, depression and burnout in 2023 compared to 2022 and a significant increase in these factors for 2024 (see Figures 1 and 2). A possible explanation for this change may be the high hopes in 2023 for a Ukrainian counteroffensive against Russia that ended with battlefield disappointment, a more somber mood among troops and anxiety about the future of Western aid for Ukraine's war effort (Bezrukova et al., 2025; PBS News, 2023).

In dire, life-threatening conditions, it is difficult to obtain timely and useful information (Institute of Medicine, 2015; Isralowitz, 2017). However, we collectively believe that this article has implications for informed decision-making related to policies and services to promote the mental health and well-being of women in war-torn environments, in Ukraine and elsewhere.

## Policy implications

The 3-year war in Ukraine has had a negative impact on the mental health of Ukrainian women. The present findings show, regardless of age, women are at higher risk of developing psychological disorders than men, and mental health complaints are most common in the 18–29 years age group, which coincides with university students. An additional source of stress is students' fear and anxiety of obtaining a quality education, as well as building a professional career under war conditions. The connection between female mental health and reproduction is another factor of concern.

The problem of preserving Ukrainian female students' mental health during the war has multiple policy and service program implications. The first involves creating a system of psychological assistance and support for students at universities, including the use of reliable and effective mental health interventions. The creation of such a system is recognized as relevant not only for Ukraine but also for other countries faced with disaster conditions. Female students tend to be more prevalent in certain study fields; therefore, curriculum and teaching methods may need to be altered to address gender-sensitive issues in study disciplines, such as education, nursing and social work (Dopelt & Houminer-Klepar, 2024; Pinchuk et al., 2025).

At the national level, war has placed a strain on the psychological and psychiatric care system. One possible way to reduce this burden may be enabling nonspecialists (e.g., representatives of NGO, religious communities and volunteers) to provide short-term psychological interventions based on teaching and training methods developed for such purposes (Findley et al., 2016; USAID, 2016; WHO, 2018; Kuryliak & Balaklytskyi, 2024; Pinchuk et al., 2024; Shraga et al., 2025). Following the experience of living under constant security threats and war (Myers-JDC-Brookdale Institute, 2023), resilience centers have begun to emerge in Ukraine to restore mental health and well-being (Pinchuk et al., 2024). The establishment of university resilience centers can be a useful contribution to strengthening and maintaining the mental health of students and university personnels.

## Limitations

The present study has limitations. The study used convenience samples obtained from a snowball method of data collection. The cross-sectional design and use of purposive samples of women limit the ability to generalize study results. However, we believe this does not lessen the significance of the study. Furthermore, the use of an online survey made it possible for only those who had access to the Internet, restricting possible survey participants because of communication and power supply failures.

**Open peer review.** To view the open peer review materials for this article, please visit http://doi.org/10.1017/gmh.2025.10112.

**Data availability statement.** The datasets analyzed during this study are available from the corresponding author upon reasonable request.

**Author contribution.** R.I. and A.R. conceptualized and designed the study. A.K., N.K. and I.P. translated and adapted study instruments into Ukrainian. A.K., N.K., I.P., V.P. and A.D. conducted data collection and administered the study. A.R. and S.R.P. conducted the statistical analyses. R.I., A.R. and S.R.P. wrote the first draft of the manuscript. A.K., N.K., I.P., V.P. and A.D. reviewed the draft and provided critical feedback. All authors approved the final version of the manuscript.

**Financial support.** This research received no specific grant from any funding agency, commercial or not-for-profit sectors.

**Competing interests.** The authors declare none.

**Ethical statement.** The study was conducted in accordance with the Declaration of Helsinki. The study was approved by the Ben Gurion University institutional review board (Approval: 22122022). Ethical review and approval were waived for this study by the Ethics Committees of the Lviv State University of Physical Culture, Ukraine; Faculty of Psychology and Natural Sciences of the Rivne State University of Humanities, Ukraine; Faculty of Psychology of the Taras Shevchenko National University of Kyiv, Institute of Psychology and Social Work, T.H. Shevchenko National University "Chernihiv Colehium," Ukraine; and Faculty of Psychology of the V. N. Karazin Kharkiv National University, due to anonymous data collection and reporting procedures used.

**Consent to participate.** The survey was conducted online and anonymously using the Qualtrics platform. Informed consent was contained in the introductory part of the questionnaire. In the case of refusal to participate, an automatic exit from the online survey system occurred. The start of the survey means respondents' informed consent.

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
