## [Reviewer Report]

The study offers a timely and critical exploration of the psychological toll of prolonged war on a vulnerable population, young women pursuing higher education in Ukraine. By analysing data collected across three distinct phases of the war, the authors provide valuable insights into evolving patterns of fear, depression, burnout, and loneliness. The study highlights the urgent need for targeted mental health interventions within higher education settings in Ukraine. It also underscores the importance of gender-sensitive approaches and sustained psychosocial support during protracted crises.

There are several minor changes I suggest implementing:

1) While describing the PHQ-9 data, it would be great to have the % prevalence of depression highlighted, also in terms of the changes from T1 to T3, based on the cut-off score. Currently, only the scores are presented, which is also important, but less informative.

2) Figures 1, 2, 3, 4 start not from the “0” point, which creates a bit of a distortion of perception. I suggest all of them start from 0.

3) It would also be helpful to mention the reporting guideline the study followed. I believe it might be the STROBE. In this case, it will also be essential to review the STROBE checklist and implement the missing elements.

4) It will also be very beneficial to see the demographic breakdowns (e.g., age, region, academic field) to assess the coverage and maybe some regional differences (if such data were collected).

5) It’s unclear whether the same individuals were followed across all three time points or if different cohorts were surveyed. Please clarify whether the study design is truly longitudinal or a repeated cross-sectional design.

6) Strengthen the implications by linking findings to specific interventions, such as university-based counselling, trauma-informed curricula, or national mental health strategies.

---

## [Reviewer Report]

I have the following comments for the authors to address. I am happy to review this paper again.

1) For Introduction, the authors provided an general literature review. The critical question remains, why they only focus on female university students? I recommend the authors to provide recent research findings from the Russo-Ukrainian war that highlights the role of female gender and discuss the following finding in the introduction (Not limited to, additional literature in the current war are welcome):

Search PubMed for: Multivariate linear regression analyses found that female gender, Ukrainian and Polish citizenship, household size, self-rating health status, past psychiatric history, and avoidance coping were significantly associated with higher DASS-21 and IES-R scores after adjustment of other variables

2) Under the methods, please state the ethicis or IRB approval. Please state the statisical analysis method.

3) Under the discussion, please discussion intervention to improve mental health, specifically related to Ukrainians during the current Russo-Ukrainian war. Please include the following evidence based intervention (volunteer work) in the discussion to improve mental health of female university students (Not Limited to, other evidence based interventions during the current war should be mentioned):

Search PubMed for: Ukrainian volunteers could significantly reduce negative feelings and strengthen social networks and religious faith by volunteering

---

## [Reviewer Report]

Thank you for this opportunity to revise the manuscript titled “Mental Health and Wellbeing among Ukrainian Female University Students: The Impact of War Over Three Years” that was submitted to Global Mental Health. I would like to appreciate you for performing the work on the important problem. The importance and pertinence of the research undertaken by the paper’s author(s) is demonstrably clear and readily apparent. Nevertheless, in my opinion, the manuscript requires a stronger and more convincing rationale for publication. It is not enough to merely report the research outcomes. The conclusions must also have significant relevance and practical value for various professionals. The list of potential beneficiaries should encompass academic researchers, clinical psychologists, and educational professionals working in universities. Unfortunately, the current version of the paper fails to adequately demonstrate this essential aspect of wide-ranging applicability and usefulness. The connection between the study’s findings and their real-world application across different professional fields remains underdeveloped.

Additionally, I leave several comments about the manuscript which are listed below:

1. Please, improve Introduction section. The introduction should state the clear contribution to the literature. I miss a clear description of gaps in the

literature that this study seeks to address.

2. You will need to update your reference with newly published articles with close and similar settings, as there is a large literature on the subject:

https://doi.org/10.1080/20008066.2024.2394296

https://doi.org/10.3390/socsci11090393

https://doi.org/10.1080/20008066.2022.2104009

https://doi.org/10.1080/13617672.2022.2158019

https://doi.org/10.1037/cdp0000522

3. The method should be rewritten in a more organized way. I think that the authors need to add a detailed argumentation.

4. In addition, It would be beneficial for the authors to provide a detailed and clear description of the methodology used for participant initial selection, including the criteria for eligibility and the process of randomization.

5. The Conclusion section must provide a clear and detailed explanation of how the research results can be practically applied in everyday situations. It should also show how these findings can enhance our knowledge and contribute to solving mental health and well-being challenges faced by female students in Ukrainian higher education institutions. This means translating theoretical conclusions into actionable strategies that can directly benefit the target audience and improve their psychological well-being.

Such an approach would significantly boost the paper’s overall usefulness and influence.

I will be glad to review the revised manuscript

---

## [Reviewer Report]

I find the paper to bring valuable insight into the dynamics of mental health states during war. The main concern is that it’s hard to compare the samples in the three time periods. Did the authors collect any demographic, socio-economic data, and times when the survey was completed to a) create a summary table for each time period to show how comparable or non-comparable the samples are, b) control for these factors when estimating the time effects?

Generally, it would be useful to know more about the sample - from which universities\ parts of Ukraine are the students, how exactly the survey was sent to them and how the procedure of reaching out differed in each time period.

---

## [Reviewer Report]

Thank you for sending this manuscript for another round.

The authors did a great job in processing the comments.

I think the paper is in good shape and can, in my opinion, be published in Cambridge Prisms: Global Mental Health.